# HETEROGENEOUS-MODAL UNSUPERVISED DOMAIN ADAPTATION VIA LATENT SPACE BRIDGING

## ABSTRACT

Unsupervised domain adaptation (UDA) methods effectively bridge domain gaps but become struggled when the source and target domains belong to entirely distinct modalities. To address this limitation, we propose a novel setting called Heterogeneous-Modal Unsupervised Domain Adaptation (HMUDA), which enables knowledge transfer between completely different modalities by leveraging a bridge domain containing unlabeled samples from both modalities. To learn under the HMUDA setting, we propose Latent Space Bridging (LSB), a specialized framework designed for the semantic segmentation task. Specifically, LSB utilizes a dual-branch architecture, incorporating a feature consistency loss to align representations across modalities and a domain alignment loss to reduce discrepancies between class centroids across domains. Extensive experiments conducted on six benchmark datasets demonstrate that LSB achieves state-of-the-art performance.

## 1 INTRODUCTION

Unsupervised domain adaptation (UDA) methods (Yang et al., 2020; Gu et al., 2022; Zhao et al., 2022) are powerful in transferring knowledge from a labeled source domain to an unlabeled target domain, particularly when the two domains exhibit significant distributional differences. While traditional UDA methods (Long et al., 2015; Li et al., 2019; Luo et al., 2019; Saltori et al., 2022; Xiao et al., 2022) have achieved notable success in scenarios where the source and target domains share the same modality (e.g., the image modality), multi-modal UDA (a.k.a. cross-modal UDA) (Jaritz et al., 2020; Peng et al., 2021; Wu et al., 2023; Zhang et al., 2022) has been recently proposed to address a more complex task of transferring knowledge from the source domain to the target one with multiple modalities (e.g., image and 3D point cloud). However, a significant challenge arises when the source and target domains belong to entirely distinct modalities, such as transferring knowledge from 2D images to 3D point clouds.

Consider the semantic segmentation task (Guo et al., 2018; Strudel et al., 2021; Li et al., 2022a) in real-world applications. In autonomous driving (Cheng et al., 2021; Tang et al., 2020; Yan et al., 2022; Zhuang et al., 2021), accurately segmenting objects in 3D point clouds captured by LiDAR sensors is essential for safe navigation. However, acquiring labeled 3D point cloud data is expensive and time-consuming (Liu et al., 2021). Conversely, large-scale 2D segmentation datasets (Cordts et al., 2016; Zhou et al., 2017; Caesar et al., 2018; Neuhold et al., 2017) are far more abundant, and annotating 2D image data is significantly easier and more cost-effective (Kirillov et al., 2023). This disparity motivates the need to transfer knowledge from labeled 2D images to improve the segmentation performance on unlabeled 3D point clouds.

Despite its importance, several challenges exist in this heterogeneous transfer (Day and Khoshgoftaar, 2017). First, the inherent differences in data structure and representation between modalities make it difficult to transfer knowledge directly (Jaritz et al., 2020; 2022; Peng et al., 2021). For example, 2D images are dense and grid-structured, while 3D point clouds are sparse and irregular. Second, the absence of labeled data in the target domain exacerbates the difficulty of learning effective representations. Existing UDA methods, which assume the same modality/modalities for both source and target domains, are ill-suited to address these challenges.

To address those issues, we formally propose a new setting called Heterogeneous-Modal Unsupervised Domain Adaptation (HMUDA), where the source domain data (e.g., 2D images) and unlabeled target domain data (e.g., 3D point clouds) belong to different modalities. As unlabeled data can be easily

obtained (Xiao et al., 2024), HMUDA assumes the existence of a bridge domain containing unlabeled samples with both source and target modalities. The HMUDA setting is visually depicted in Figure 1, and Table 1 provides a comparative analysis, highlighting its differences from existing domain adaptation (DA) paradigms, including UDA, multi-modal UDA (MM-UDA), and heterogeneous domain adaptation (HDA) (Fang et al., 2022; Wang and Mahadevan, 2011). Then, we propose Latent Space Bridging (LSB), a novel HMUDA framework specifically designed for semantic segmentation tasks. Specifically, the proposed LSB method employs a dual-branch architecture, comprising a source network and a target network tailored for the source and target modalities, respectively. Those networks are trained to perform pointwise segmentation using the source data with ground truth labels and the bridge data with pseudo labels. To enhance the feature alignment, we propose a **feature consistency loss** to encourage similar feature representations for samples with both modalities in the bridge domain and a **domain alignment loss** to minimize discrepancies between class centroids in the source and target domains. Experimental results across various benchmark datasets demonstrate that the proposed LSB method effectively transfers knowledge from the source domain to the target domain, outperforming both the source-only method and existing UDA methods by a significant margin.

Our contributions are summarized as follows.

- We introduce a new DA setting, heterogeneous-modal unsupervised domain adaptation, to facilitate knowledge transfer between heterogeneous modalities.
- We propose the Latent Space Bridging method, a tailored solution for the semantic segmentation task under the HMUDA setting.
- Extensive experiments on benchmark datasets demonstrate the effectiveness of the proposed LSB method, showcasing its ability to outperform existing methods.

## 2 RELATED WORK

### 2.1 UNSUPERVISED DOMAIN ADAPTATION

Unsupervised domain adaptation (UDA) (Zhuang et al., 2020) aims to transfer knowledge from a labeled source domain to an unlabeled target domain, addressing the challenge of the domain shift. Traditional UDA methods (Long et al., 2015; Sun and Saenko, 2016; Tzeng et al., 2014; Ganin et al., 2016) have achieved significant success in various applications by aligning the feature distributions between the source and target domains. For instance, discrepancy-based methods such as DAN (Long et al., 2015) and CORAL (Sun and Saenko, 2016) minimize the statistical distance between source and target feature distributions. Meanwhile, adversarial-based methods like DANN (Ganin et al., 2016) and ADDA (Tzeng et al., 2017) employ adversarial training to learn domain-invariant features, leveraging a domain classifier to ensure that extracted features cannot be distinguished as originating from either the source or target domain.

In semantic segmentation, UDA methods have been extended to tackle the pixel-level alignment challenge. Many of these methods also employ adversarial training for aligning the two domains. For instance, CyCADA (Hoffman et al., 2018) uses cycle-consistent adversarial networks to adapt both the pixel and feature levels. AdaptSegNet (Tsai et al., 2018) incorporates adversarial training at the output space to align the segmentation maps. Recently, CLAN (Luo et al., 2019) and CoSMix (Saltori et al., 2022) further refine the adversarial approach by focusing on class-level alignment and contextual consistency. However, adversarial training can be unstable and a high computational burden for segmentation tasks (Mo et al., 2022). Different from them, the proposed LSB method uses discrepancy-based feature alignment, which directly minimizes the difference between source and target feature distributions.

### 2.2 MULTI-MODAL DOMAIN ADAPTATION

Multi-modal domain adaptation (Jaritz et al., 2020; Peng et al., 2021; Zhang et al., 2022; Wu et al., 2024) involves transferring knowledge across domains with multiple modalities, such as images and text, or images and 3D point clouds. This approach leverages the complementary information from different modalities to enhance the adaptation performance. For example, Jaritz et al. introduce xMUDA (Jaritz et al., 2020), a cross-modal learning method that combines RGB images and LiDAR

point clouds for improving 3D semantic segmentation accuracy. DsCML (Peng et al., 2021) employs adversarial learning at the output level to model domain-invariant representations. SSE-xMUDA (Zhang et al., 2022) presents a self-supervised exclusive learning mechanism that exploits the unique information of different modalities to complement each other. Dual-Cross (Li et al., 2022b) designs a multi-modal stylized transfer module to alleviate the domain shift problem. These existing methods assume that both the source and target domains contain multi-modal data, and typically, adaptation only occurs between the same type of modal data. In contrast, our approach introduces a novel setting where both the source and target domains contain only one type of modality. To bridge the modal gap, we introduce an unlabeled bridge domain that possesses both source and target modalities.

## 2.3 HETEROGENEOUS DOMAIN ADAPTATION

Heterogeneous domain adaptation (HDA) addresses the adaptation between domains with different feature spaces or data types, presenting unique challenges due to inherent differences in feature representations. HDA methods can be divided into two categories: symmetric transformations (Duan et al., 2012; Zhang et al., 2017; Samat et al., 2017; Wang and Mahadevan, 2011; Tsai et al., 2016) and asymmetric transformations (Feuz and Cook, 2015; Zhou et al., 2014b; Nam and Kim, 2015; Zhou et al., 2014a). Symmetric transformation methods, like HFA (Duan et al., 2012) and JGSA (Zhang et al., 2017), involve projecting both source and target domains into a common latent space for alignment. This approach facilitates a balanced representation by leveraging complementary information from both domains. In contrast, asymmetric transformation methods (Feuz and Cook, 2015) focus on transforming one domain to align with the other. Although effective, most HDA methods require partial target domain labels for guiding the adaptation process (Feuz and Cook, 2015; Duan et al., 2012), especially when aligning feature spaces (Xiao and Guo, 2014). Moreover, the majority of HDA methods are not end-to-end, which necessitates separate stages for feature extraction and alignment (Wang and Mahadevan, 2011; Wang and Breckon, 2022). In contrast, our approach is both end-to-end and label-free in the target domain, simultaneously training networks for both modalities.

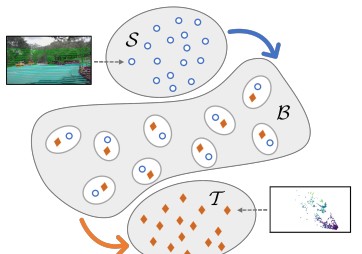

Figure 1: The illustration of HMUDA setting.

Table 1: The comparison between HMUDA and other DA settings. The $\mathcal{M}_1$ and $\mathcal{M}_2$ represent different modalities, 'E-to-E' indicates whether the corresponding setting uses the end-to-end training strategy, and 'Ex-Label' indicates whether the corresponding setting uses extra labeled target data.

|  | Source | | Target | | E-to-E | Ex-Label |
|---|---|---|---|---|---|---|
|  | $\mathcal{M}_1$ | $\mathcal{M}_2$ | $\mathcal{M}_1$ | $\mathcal{M}_2$ | | |
| UDA | ✓ | ✗ | ✓ | ✗ | ✓ | ✗ |
| MM-UDA | ✓ | ✓ | ✓ | ✓ | ✓ | ✗ |
| HDA | ✓ | ✗ | ✗ | ✓ | ✗ | ✓ |
| HMUDA | ✓ | ✗ | ✗ | ✓ | ✓ | ✗ |

## 3 PROBLEM FORMULATION FOR HMUDA

In this section, we introduce the HMUDA setting. Under the HMUDA setting, there exists a labeled source domain $\mathcal{S} = \{(\mathbf{x}^s, \mathbf{y}^s)\}$, where each sample consists of an input $\mathbf{x}^s \in \mathcal{M}_1$ and pointwise segmentation labels $\mathbf{y}^s \in \mathbb{R}^{C \times N}$ for $\mathbf{x}^s$. Here, $\mathcal{M}_1$ denotes the source modality of the input (e.g., the 2D image), $N$ is the number of labeled points in the input for semantic segmentation, and $C$ is the number of classes in the semantic segmentation task. Moreover, there exists a unlabeled target domain $\mathcal{T} = \{\mathbf{x}^t\}$, where $\mathbf{x}^t$ belongs to a different modality $\mathcal{M}_2$ (e.g., the 3D point cloud). Additionally, we assume the existence of an unlabeled bridge domain $\mathcal{B} = \{(\mathbf{x}^{bs}, \mathbf{x}^{bt})\}$, where $\mathbf{x}^{bs} \in \mathcal{M}_1$ and $\mathbf{x}^{bt} \in \mathcal{M}_2$ are from the two modalities and correspond to the same input. Note that the bridge domain could differ from the source and target domains. Under the HMUDA setting, we aim to transfer the knowledge in $\mathcal{S}$ to help the learning of $\mathcal{T}$ with the help of the bridge domain $\mathcal{B}$. In the following, we give the definition for HMUDA.

**Definition 1.** *(HMUDA) Heterogeneous Modality Unsupervised Domain Adaptation (HMUDA) transfers knowledge from a labeled source domain $\mathcal{S}$ to an unlabeled target domain $\mathcal{T}$ across different modalities. This transfer is facilitated by an unlabeled bridge domain $\mathcal{B}$, which provides paired samples from both source and target modalities. The objective of HMUDA is to leverage the labeled*

Figure 2: An illustration of the proposed LSB framework. $\mathcal{S}$, $\mathcal{T}$, and $\mathcal{B}$ denote the source, target, and bridge domains, respectively. $\mathbf{x}^s$, $\mathbf{x}^{bs}$ and $\mathbf{x}^{bt}$ represent samples from each corresponding domain. Lines within different colors denote the data flow for computing different losses.
*data from $\mathcal{S}$ and the paired data in $\mathcal{B}$ to improve the learning and performance on the target domain $\mathcal{T}$.*

As shown in Table 1, HMUDA differs from existing DA settings. Specifically, unsupervised domain adaptation (UDA) addresses the domain gap between source and target domains within the same modality. Multi-modal UDA (MM-UDA) assumes the existence of different modalities for both source and target domains. Heterogeneous domain adaptation (HDA) transfers knowledge from the source domain to the target domain with homogeneous or heterogeneous modalities but usually requires extra labeled target data. Moreover, HDA manually extracts features from the original data without learning the feature extractors and hence it does not support end-to-end training. Compared with UDA and MM-UDA, HMUDA considers a more complex scenario involving heterogeneous source and target modalities. Different from HDA, HMUDA supports the end-to-end training without requiring additional labeled target data, making it more streamlined and practical than HDA.

## 4 METHODOLOGY

In this section, we present the proposed LSB method for HMUDA. We first introduce the entire architecture of the proposed LSB method in the architecture section. Then, we introduce the losses in LSB to learn to bridge the source and target model by the bridge domain and align class centroids across domains. Next, we give the entire objective function and the algorithm for the LSB method. Finally, we provide a theoretical analysis of the error bound under the HMUDA setting.

### 4.1 ARCHITECTURE

As depicted in Figure 2, the proposed LSB method employs a dual-branch architecture to predict pointwise segmentation labels. The architecture in the LSB method consists of two distinct segmentation networks: (i) a **source network** $\{h, f\}$ tailored for the source modality $\mathcal{M}_1$ with a feature extractor $h(\cdot) : \mathcal{M}_1 \rightarrow \mathbb{R}^{d_h \times N}$ and a classifier $f(\cdot) : \mathbb{R}^{d_h \times N} \rightarrow \mathbb{R}^{C \times N}$, where $d_h$ is the feature dimension of $h$. (ii) a **target network** $\{\phi, g\}$ designed for the target modality $\mathcal{M}_2$ with a feature extractor $\phi(\cdot) : \mathcal{M}_2 \rightarrow \mathbb{R}^{d_\phi \times N}$ and a classifier $g(\cdot) : \mathbb{R}^{d_\phi \times N} \rightarrow \mathbb{R}^{C \times N}$, where $d_\phi$ is the feature dimension of $\phi$.

During training, the source domain and bridge domain data associated with modality $\mathcal{M}_1$ are input into the source network, while the target and bridge domain data belonging to modality $\mathcal{M}_2$ are input into the target network. Both the source and target networks can predict the pointwise segmentation labels independently. For example, given a target input $\mathbf{x}^t$, the target model generates the probability distribution of $N$ points over $C$ classes as $g(\phi(\mathbf{x}^t))$. Further implementation details are provided in the experimental setting section.

### 4.2 BRIDGING SOURCE AND TARGET NETWORKS

To train the source network, a natural solution is to minimize the cross-entropy loss between the prediction of input $\mathbf{x}^s$ (i.e., $f(h(\mathbf{x}^s))$) and its corresponding labels $\mathbf{y}^s$. This segmentation loss, denoted by $\mathcal{L}_{\text{seg}}^s$, is formally defined as

$$\mathcal{L}_{\text{seg}}^s(\mathbf{x}^s, \mathbf{y}^s) = -\frac{1}{N} \sum_{i=1}^{N} (\mathbf{y}_{:,i}^s)^\top \log(f(h(\mathbf{x}^s))), \tag{1}$$

where $\mathbf{y}_{:,i}^s$ represents the $i$-th column of $\mathbf{y}^s$ corresponding to the one-hot label for the $i$-th segment point.

Similarly, one can train the target network using the segmentation loss based on the target input $\mathbf{x}^t$ and its corresponding labels. However, since labels for the target domain samples are unavailable, directly training the target network becomes infeasible. To address this issue, we use the bridge domain instead to train the target network. Specifically, we introduce a teacher source model, comprising $\hat{f}$ and $\hat{h}$, which are updated from $f$ and $h$ using the exponential moving average (EMA) (Tarvainen and Valpola, 2017) at each iteration as

$$\boldsymbol{\theta}_{\hat{f}} \leftarrow \alpha\boldsymbol{\theta}_{\hat{f}} + (1-\alpha)\boldsymbol{\theta}_f, \ \boldsymbol{\theta}_{\hat{h}} \leftarrow \alpha\boldsymbol{\theta}_{\hat{h}} + (1-\alpha)\boldsymbol{\theta}_h, \tag{2}$$

where $\boldsymbol{\theta}_f$, $\boldsymbol{\theta}_h$, $\boldsymbol{\theta}_{\hat{f}}$ and $\boldsymbol{\theta}_{\hat{h}}$ denotes the parameters of $f$, $h$, $\hat{f}$, $\hat{h}$, respectively. $\alpha$ is a hyperparameter that is dynamically adjusted during training as

$$\alpha = \min\left(1 - \frac{1}{t+1}, \alpha\right), \tag{3}$$

where $t$ denotes the number of training iterations completed so far. The pseudo-label $\hat{\mathbf{y}}^b$ for the sample pair $(\mathbf{x}^{bs}, \mathbf{x}^{bt})$ in the bridge domain $\mathcal{B}$ is generated by the teacher source network as

$$\hat{\mathbf{y}}^b = \text{one-hot}\left(\arg\max \hat{f}(\hat{h}(\mathbf{x}^{bs}))\right), \tag{4}$$

where one-hot$(\cdot)$ denotes the transformation that converts the prediction into the one-hot vector. Then, we propose the loss $\mathcal{L}_{\text{seg}}^b$ to train the target network on the bridge domain $\mathcal{B}$ as

$$\mathcal{L}_{\text{seg}}^b(\mathbf{x}^{bs}, \mathbf{x}^{bt}) = -\frac{1}{N}\sum_{i=1}^{N}(\hat{\mathbf{y}}_{:,i}^b)^{\top}\log(g(\phi(\mathbf{x}^{bt}))), \tag{5}$$

where $\hat{\mathbf{y}}_{:,i}^b$ denotes the $i$-th column of $\hat{\mathbf{y}}^b$ corresponding to the one-hot pseudo label for the $i$-th segment point.

For an input sample pair $(\mathbf{x}^{bs}, \mathbf{x}^{bt})$, the source and target networks should extract similar features since they share the same labels. To encourage the consistent features between the source and target networks, we introduce learnable projections $p_h : \mathbb{R}^{d_h \times N} \to \mathbb{R}^{d \times N}$ and $p_\phi : \mathbb{R}^{d_\phi \times N} \to \mathbb{R}^{d \times N}$, which map the source and target feature into a $d$-dimensional shared feature space, respectively. To minimize the discrepancy between the projected features of the source and target networks, we define the feature consistency loss $\mathcal{L}_{\text{con}}^b$ as

$$\mathcal{L}_{\text{con}}^b(\mathbf{x}^{bs}, \mathbf{x}^{bt}) = \frac{1}{N}\sum_{i=1}^{N}\left(\lambda_w||\mathbf{w}||_2^2 + ||p_h(h(\mathbf{x}^{bs})) - p_\phi(\phi(\mathbf{x}^{bt}))||_2^2\right) \tag{6}$$

where $||\cdot||_2$ denotes the $\ell_2$ norm of a vector and $\lambda_w > 0$ is a hyper-parameter controlling the strength of the regularization term $||\mathbf{w}||_2^2$, which is applied to parameters in $p_h$ and $p_\phi$.

### 4.3 CROSS-MODAL DOMAIN ALIGNMENT

Directly training the models using the source and bridge domain often leads to overfitting(Tzeng et al., 2014), resulting in diminished performance on the target domain. To alleviate this problem, inspired by the discrepancy-based UDA methods (Long et al., 2015; Zhu et al., 2021), we learn the representation that minimizes the distance between the source and target domains. To begin with, we obtain the pseudo label $\hat{\mathbf{y}}^t$ for each $\mathbf{x}^t \in \mathcal{T}$ by the target network. For class $c$, we define the class centroid features $\mathbf{m}_c^s \in \mathbb{R}^d$ for the source domain and $\mathbf{m}_c^t \in \mathbb{R}^d$ for the target domain as

$$\mathbf{m}_c^s = \left(\sum_{(\mathbf{x}^s, \mathbf{y}^s) \in \mathcal{S}} \mathbf{y}_c^s \mathbf{1}\right)^{-1} \sum_{(\mathbf{x}^s, \mathbf{y}^s) \in \mathcal{S}} p_h(h(\mathbf{x}^s))(\mathbf{y}_c^s)^{\top} \tag{7}$$

$$\mathbf{m}_c^t = \left(\sum_{\mathbf{x}^t \in \mathcal{T}} \hat{\mathbf{y}}_c^t \mathbf{1}\right)^{-1} \sum_{\mathbf{x}^t \in \mathcal{T}} p_\phi(\phi(\mathbf{x}^t))(\hat{\mathbf{y}}_c^t)^{\top}, \tag{8}$$

where $\mathbf{y}_c^s$ and $\hat{\mathbf{y}}_c^t$ denote the $c$-th row element of $\mathbf{y}^s$ and $\hat{\mathbf{y}}^t$, respectively, and $\mathbf{1}$ denotes an $N$ dimension vector where all elements equal to one. We expect the source and target models to share similar centroid features for each class. To this end, we minimize the discrepancy between the centroids using the alignment loss $\mathcal{L}_{\mathrm{ali}}$ as

$$\mathcal{L}_{\mathrm{ali}}(\mathcal{S}, \mathcal{T}) = \frac{1}{C} \sum_{c=1}^{C} 1 - \cos\left(\mathbf{m}_c^s, \mathbf{m}_c^t\right), \tag{9}$$

where $\cos(\cdot)$ denotes the cosine similarity function.

## 4.4 OBJECTIVE FUNCTION AND ALGORITHM

We jointly learn the source and target networks by minimizing the final objective $\mathcal{L}(\mathcal{S}, \mathcal{B}, \mathcal{T})$, which combines $\mathcal{L}_{\mathrm{seg}}^s$, $\mathcal{L}_{\mathrm{seg}}^b$, $\mathcal{L}_{\mathrm{con}}$ and $\mathcal{L}_{\mathrm{ali}}$ together, i.e.,

$$\begin{aligned}
\mathcal{L}(\mathcal{S}, \mathcal{B}, \mathcal{T}) = \sum_{(\mathbf{x}^s, \mathbf{y}^s) \in \mathcal{S}} &\left(\mathcal{L}_{\mathrm{seg}}^s(\mathbf{x}^s, \mathbf{y}^s)\right) + \lambda_{\mathrm{a}} \mathcal{L}_{\mathrm{ali}}(\mathcal{S}, \mathcal{T}) \\
&+ \sum_{(\mathbf{x}^{bs}, \mathbf{x}^{bt}) \in \mathcal{B}} \left(\mathcal{L}_{\mathrm{seg}}^b(\mathbf{x}^{bs}, \mathbf{x}^{bt}) + \lambda_{\mathrm{c}} \mathcal{L}_{\mathrm{con}}^b(\mathbf{x}^{bs}, \mathbf{x}^{bt})\right),
\end{aligned} \tag{10}$$

where $\lambda_{\mathrm{c}} > 0$ and $\lambda_{\mathrm{a}} > 0$ are hyperparameters to balance different losses. The overall algorithm for the LSB method is provided in the Appendix.

## 4.5 THEORETICAL ANALYSIS

In this section, we analyze the error bound of HMUDA. Let $\mathcal{E}_s(\{h, f\}) = \mathbb{E}_{(\mathbf{x}, \mathbf{y}) \sim \mathcal{D}_s}[f(h(\mathbf{x})) \neq \mathbf{y}]$ and $\mathcal{E}_t(\{\phi, g\}) = \mathbb{E}_{(\mathbf{x}, \mathbf{y}) \sim \mathcal{D}_t}[g(\phi(\mathbf{x})) \neq \mathbf{y}]$ denote the expected error in the source domain with data distribution $\mathcal{D}_s$ and target domain with data distribution $\mathcal{D}_t$, respectively. For bridge domain distribution $\mathcal{D}_b$, let $\mathcal{E}_{bs}(\{h, f\}) = \mathbb{E}_{(\mathbf{x}^{bs}, \mathbf{x}^{bt}, \mathbf{y}) \sim \mathcal{D}_b}[f(h(\mathbf{x}^{bs})) \neq \mathbf{y}]$ and $\mathcal{E}_{bt}(\{\phi, g\}) = \mathbb{E}_{(\mathbf{x}^{bs}, \mathbf{x}^{bt}, \mathbf{y}) \sim \mathcal{D}_b}[g(\phi(\mathbf{x}^{bt})) \neq \mathbf{y}]$ be the errors corresponding to the source and target modalities within the bridge domain. We denote the hypothesis space for the source network $\{h, f\}$ and target network $\{\phi, g\}$ as $\mathcal{H}_s$ and $\mathcal{H}_t$, respectively. The optimized source and target networks are defined as follows.

**Definition 2.** *The ideal joint hypothesis for source and target modality is the hypothesis that minimizes the combined errors:*

$$\{h^*, f^*\} = \underset{\{h, f\} \in \mathcal{H}_s}{\arg\min} \; \mathcal{E}_s(\{h, f\}) + \mathcal{E}_{bs}(\{h, f\}); \tag{11}$$

$$\{\phi^*, g^*\} = \underset{\{\phi, g\} \in \mathcal{H}_t}{\arg\min} \; \mathcal{E}_{bt}(\{\phi, g\}) + \mathcal{E}_t(\{\phi, g\}). \tag{12}$$

The combined error for source and target modality under the ideal hypothesis are denoted as:

$$\begin{aligned}
\lambda_s &= \mathcal{E}_s(\{h^*, f^*\}) + \mathcal{E}_{bs}(\{h^*, f^*\}); \\
\lambda_t &= \mathcal{E}_{bt}(\{\phi^*, g^*\}) + \mathcal{E}_t(\{\phi^*, g^*\}).
\end{aligned} \tag{13}$$

Moreover, similar to (Zhuang et al., 2024), we assume the error gap between modalities of the same domain is bounded by their feature gap as $\left|\mathcal{E}_{bt}(\{\phi, g\}) - \mathcal{E}_{bs}(\{h, f\})\right| \leq L\mathbb{E}_{(\mathbf{x}^{bs}, \mathbf{x}^{bt}) \sim \mathcal{D}_b}\left(d(h(\mathbf{x}^{bs}), \phi(\mathbf{x}^{bt}))\right)$, where $L$ is a constant and $d$ is the distance function (e.g., the $\ell_1$ distance). We are now ready to give a bound on the expected error of the target domain in the following theorem.[1]

**Theorem 1.** *For every $\{\phi, g\} \in \mathcal{H}_t$, the target domain error is bounded as:*

$$\begin{aligned}
\mathcal{E}_t(\{\phi, g\}) \leq \mathcal{E}_s(\{h, f\}) &+ L\mathbb{E}_{(\mathbf{x}^{bs}, \mathbf{x}^{bt}) \sim \mathcal{D}_b}\left(d(h(\mathbf{x}^{bs}), \phi(\mathbf{x}^{bt}))\right) \\
&+ \frac{1}{2} d_{\mathcal{H}_s \Delta \mathcal{H}_s}(\mathcal{D}_s, \mathcal{D}_b) + \frac{1}{2} d_{\mathcal{H}_t \Delta \mathcal{H}_t}(\mathcal{D}_b, \mathcal{D}_t), \\
&+ (\lambda_s + \lambda_t)
\end{aligned} \tag{14}$$

---

[1]The proof can be found in the Appendix.

Table 2: Testing mIoU results on HMUDA tasks. The best performance is in **bold**.

| | | $USA \to Sing.$ | $Day \to Night$ | $Virt. \to A2D2$ | $USA \to Sing.$ | $Day \to Night$ | $Virt. \to Sem.$ | $Sem. \to A2D2$ | $A2D2 \to Sem.$ |
|---|---|---|---|---|---|---|---|---|---|
| | Bridge Domain $\mathcal{B}$ | | Sem. | | | A2D2 | | Virt. | |
| **2D→3D** | Oracle | 77.69 | 73.71 | 71.50 | 77.69 | 73.71 | 82.75 | 71.50 | 82.75 |
| | xMUDA | 65.12 | 75.11 | 61.03 | 65.12 | 75.11 | 57.97 | 44.95 | 65.07 |
| | Source-Only | 51.04 | 57.32 | 19.00 | 49.21 | 55.96 | 36.44 | 43.75 | 43.28 |
| | PL | 52.08 | 59.33 | 16.84 | 55.39 | 61.95 | **46.49** | 39.66 | 38.25 |
| | CDSPP | 17.61 | 21.81 | 9.74 | 17.61 | 21.81 | 12.44 | 11.27 | 11.00 |
| | LSB | **56.41** | **62.25** | **33.37** | **57.13** | **63.15** | 45.43 | **46.34** | **47.22** |
| **3D→2D** | Oracle | 76.28 | 62.63 | 85.67 | 76.28 | 62.63 | 87.19 | 85.67 | 87.19 |
| | xMUDA | 65.88 | 63.63 | 62.07 | 65.88 | 63.63 | 52.98 | 63.54 | 60.51 |
| | Source-Only | 43.80 | 17.69 | 42.56 | 38.99 | 25.02 | **42.22** | 31.48 | 20.06 |
| | PL | 43.89 | 26.55 | 43.37 | 34.97 | 22.67 | 38.75 | 14.45 | 34.27 |
| | CDSPP | 22.18 | 17.75 | 20.53 | 22.18 | 17.75 | 15.87 | 17.58 | 13.11 |
| | LSB | **45.35** | **33.81** | **47.92** | **39.83** | **38.36** | 38.03 | **32.49** | **39.89** |

*where $d_{\mathcal{H}_s \Delta \mathcal{H}_s}$, $d_{\mathcal{H}_t \Delta \mathcal{H}_t}$ are the $\mathcal{H} \Delta \mathcal{H}$-distance (Ben-David et al., 2010) between domains in the source and target modality, respectively.*

Theorem 1 shows that the target domain error is upper-bounded by the summation of five terms: (i) the source domain error $\mathcal{E}_s(\{h, f\})$; (ii) the modality discrepancy in the bridge domain; (iii) the ideal combined errors, which are a constant; (iv) the domain discrepancy between the source and bridge domains; (v) The domain discrepancy between bridge and target domains. In LSB, term (i) is optimized by using the segmentation loss defined in Eq. (1). Notably, term (ii) directly aligns with Eq. (6), which minimizes the feature gap between modalities. Moreover, Eq. (5) assesses term (iv), and Eq. (9) measures both terms (iv) and (v). Hence, the design of the LSB method aligns with the generalization bound in Theorem 1, which gives theoretical support for the proposed LSB method.

## 5 EXPERIMENTS

In this section, we empirically evaluate the proposed LSB method under the HMUDA setting.

### 5.1 EXPERIMENTAL SETTINGS

**Datasets.** We conduct experiments on several publicly available multimodal datasets, including (i) nuScenes-lidarseg (Caesar et al., 2020), which is divided into different scene layouts (i.e., *USA* and Singapore (*Sing.*)) and lighting conditions (i.e., *Day* and *Night*). (ii) *A2D2* (Geyer et al., 2020), which consists of data collected from Audi, featuring diverse driving scenarios with multi-sensor data. (iii) SemanticKITTI (*Sem.*) (Behley et al., 2019), a large-scale dataset providing dense pointwise semantic annotations for LiDAR scans, capturing urban environments in various driving conditions. (iv) VirtualKITTI (*Virt.*) (Gaidon et al., 2016), a synthetic dataset generated from realistic 3D simulations with precise ground truth annotations. To evaluate the proposed method under HMUDA, we construct six transfer tasks under the HMUDA setting, including: (i) *USA→Sing.* for changes in scene layout. (ii) *Day→Night* for changes in light conditions. (iii) *Virt→A2D2* and (iv) *Virt→Sem.* for synthetic to real data. (v) *Sem.→A2D2* and (vi) *A2D2→Sem.* for different camera setups. For the (i-iv) tasks, we use *Sem.* and *A2D2* as the bridge domain, while for the (v) and (vi) tasks, *Virt.* is used instead. In all datasets, the LiDAR and RGB cameras are synchronized and calibrated. Following (Jaritz et al., 2022), we only use the front camera's images in all datasets for consistency. To evaluate the generality of HMUDA, we also perform experiments on the reverse transfer task, from 3D point clouds to 2D images. Further dataset details are provided in the Appendix.

**Implementation Details.** By following (Jaritz et al., 2022), for the 2D network, we use a U-Net (Ronneberger et al., 2015) with a ResNet34 (He et al., 2016) encoder pre-trained on ImageNet (Deng et al., 2009). For the 3D network, we use the SparseConvNet(Graham et al., 2018) and U-Net(Ronneberger et al., 2015) architectures with a voxel size of 5cm. Furthermore, we use a linear layer for projections $p_\phi$ and $p_h$, respectively. Training is conducted using the Adam optimizer with a batch size of 16, an initial learning rate of 0.001, $\beta_1 = 0.9$, and $\beta_2 = 0.999$. All the parameters are trained for 50,000 steps, with a learning rate scheduler following (Jaritz et al., 2020). The hyper-paramters $\alpha$ is initially set to 0.999. The parameters $\lambda_w$, $\lambda_c$, and $\lambda_a$ are set to 0.01, 4.0, and 0.1, respectively. We use the mean Intersection over Union (mIoU) to evaluate the performance. All experiments are conducted on an NVIDIA V100 (32GB) GPU.

Table 3: Effect of losses $\mathcal{L}_{seg}^s$, $\mathcal{L}_{seg}^t$, $\mathcal{L}_{con}^b$, and $\mathcal{L}_{ali}$ in terms of mIoU for 2D-to-3D HMUDA tasks. The best performance is in **bold**.

| $\mathcal{L}_{seg}^s$ | $\mathcal{L}_{seg}^b$ | $\mathcal{L}_{con}^b$ | $\mathcal{L}_{ali}$ | $\mathcal{B}$ | USA → Sing. | Day → Night | Virt. → A2D2 | $\mathcal{B}$ | Sem. → A2D2 |
|---|---|---|---|---|---|---|---|---|---|
| ✓ | ✓ | ✗ | ✗ |  | 53.62 | 61.86 | 22.72 |  | 42.47 |
| ✓ | ✓ | ✓ | ✗ | Sem. | 52.68 | 61.12 | 28.24 | Virt. | 45.27 |
| ✓ | ✓ | ✗ | ✓ |  | 54.34 | 60.86 | 27.96 |  | **46.37** |
| ✓ | ✓ | ✓ | ✓ |  | **56.41** | **62.25** | **33.37** |  | 46.34 |
| $\mathcal{L}_{seg}^s$ | $\mathcal{L}_{seg}^b$ | $\mathcal{L}_{con}^b$ | $\mathcal{L}_{ali}$ | $\mathcal{B}$ | USA → Sing. | Day → Night | Virt. → Sem. | $\mathcal{B}$ | A2D2 → Sem. |
| ✓ | ✓ | ✗ | ✗ |  | 49.35 | 60.64 | 35.74 |  | 42.29 |
| ✓ | ✓ | ✓ | ✗ | A2D2 | 50.50 | 62.72 | 36.53 | Virt. | 42.46 |
| ✓ | ✓ | ✗ | ✓ |  | 54.49 | 62.87 | 38.55 |  | 46.95 |
| ✓ | ✓ | ✓ | ✓ |  | **57.13** | **63.15** | **45.43** |  | **47.22** |

**Baselines.** We compare the proposed LSB method against the following representative baselines. (i) **Oracle**, which trains the target network $(g, \phi)$ using segmentation loss directly on the labeled target domain $\{(\mathbf{x}^t, \mathbf{y}^t)\}$. Each input $\mathbf{x}^t$ is paired with its ground-truth label $\mathbf{y}^t$. This serves as an upper bound for HMUDA. (ii) **xMUDA** (Jaritz et al., 2022), a multimodal UDA method that jointly learns from paired 2D-3D data in both the source and target domains by enforcing cross-modal consistency through mutual mimicking. Such paired supervision provides stronger guidance than unimodal settings but requires costly annotations. We therefore regard xMUDA as a *soft upper-bound* baseline for HMUDA. (iii) **Source-Only**, where the source network is trained solely on the source domain $\mathcal{S}$, and pseudo-labels generated on the bridge domain $\mathcal{B}$ are then used to supervise the target network. (iv) **Pseudo-Labeling (PL)** (Li et al., 2019), a unimodal two-stage pseudo-labeling strategy, implemented following Jaritz et al. (2022), in which pseudo-labels produced by the source network are used to guide training of the target model. (v) **CDSPP** (Wang and Breckon, 2022), a HDA method that transfers knowledge from extracted features by learning domain-specific projections to align source and target features in a shared subspace while preserving class structure, which we reproduce using approximately 5% labeled target samples in a semi-supervised setup. For a fair comparison, we use the same backbone ResNet34 and SparseConvNet for 2D and 3D samples as LSB to extract features.

## 5.2 Main Results

Table 2 shows the testing mIoU results on HMUDA tasks. As can be seen, the LSB achieves the highest mIoU across all tasks except for *Virt.→Sem.*. For instance, on the 2D-to-3D task *Virt.→A2D2* with bridge domain *Sem.*, LSB surpasses Source-Only, PL, and CDSPP by a large margin of 14.37, 16.53, and 23.63, validating its effectiveness in addressing the heterogeneous domain gaps. LSB achieves average mIoU improvements of 5.16 and 7.10 over PL across all the 2D-to-3D and 3D-to-2D tasks, respectively, demonstrating its capability to bridge source and target networks through joint training. Moreover, LSB consistently outperforms CDSPP across all tasks, achieving average mIoU gains of 36.00 and 21.09 on the 2D-to-3D and 3D-to-2D settings, respectively, demonstrating the effectiveness of the proposed end-to-end training framework. Compared with the supervision-heavy xMUDA, which leverages paired 2D–3D data in both source and target domains, LSB achieves a higher mIoU on the 2D-to-3D transfer task *Sem.→A2D2* with the bridge domain *Virt.*, suggesting that the bridge domain can be effective for knowledge transfer even with less direct supervision. Despite these improvements, all methods under HMUDA still fall short of the Oracle, underscoring the challenges inherent in HMUDA tasks.

## 5.3 Ablation Studies

**Effect of Different Losses $\mathcal{L}_{seg}^s$, $\mathcal{L}_{seg}^t$, $\mathcal{L}_{con}^b$, and $\mathcal{L}_{ali}$.** We conduct experiments on HMUDA tasks to study the effect of losses $\mathcal{L}_{seg}^s$, $\mathcal{L}_{seg}^t$, $\mathcal{L}_{con}^b$, and $\mathcal{L}_{ali}$ w.r.t. mIoU. Specifically, we consider four combinations: (i) with the two segmentation losses $\mathcal{L}_{seg}^s$, $\mathcal{L}_{seg}^b$ only (i.e., LSB (w/ seg only)); (ii) without the alignment loss $\mathcal{L}_{ali}$ (i.e., LSB (w/o $\mathcal{L}_{ali}$)); (iii) without the consistent loss $\mathcal{L}_{con}^b$ (i.e., LSB (w/o $\mathcal{L}_{con}^b$)); (iv) with all the proposed losses $\mathcal{L}_{seg}^s$, $\mathcal{L}_{seg}^b$, $\mathcal{L}_{con}^b$ and $\mathcal{L}_{ali}$ together (i.e. the proposed LSB). Since the two segmentation losses (i.e., $\mathcal{L}_{seg}^s$ and $\mathcal{L}_{seg}^b$) are necessary for training the source and target model, we do not remove them in any variant of HMUDA.

Table 3 shows the testing mIoU results on the 2D-to-3D HMUDA tasks for different variants of LSB. As can be seen, LSB achieves the best mIoU across all HMUDA tasks, except for *Sem.→A2D2*. LSB

outperforms LSB (w/o $\mathcal{L}_{con}^b$) by a large margin of 3.97 on average, validating that encouraging the consistent features benefits the learning of the target domain. Compared with LSB (w/o $\mathcal{L}_{ali}$, LSB achieves an average improvement of 2.36, showing the effectiveness of minimizing the distance of class centriod features between source and target domain. Moreover, LSB (w/ seg only) surpasses the previous SOTA UDA method (PL in Table 2) by 1.19 on average, demonstrating the superiority of the joint training strategy.

**Effect of $p_h$ and $p_\phi$.** In LSB, we introduce two learnable projections to map the source and target features into a $d$-dimensional shared feature space. To study the effect of these two projections, we compare LSB with its variants: (i) LSB (w/o $p_h$), which uses a linear layer to map the target feature into the $d_h$-dimensional source feature space. (ii) LSB (w/o $p_\phi$), which uses a linear layer to map the source feature into the $d_\phi$-dimensional target feature space.

Table 4: Effect of $p_\phi$ and $p_h$ on three 2D-to-3D HMUDA tasks.

| Method | $\mathcal{B}$ | USA → Sing. | Day → Night | $\mathcal{B}$ | A2D2 → Sem. |
|---|---|---|---|---|---|
| LSB (w/o $p_\phi$) | | 20.78 | 3.82 | | 15.00 |
| LSB (w/o $p_h$) | A2D2 | 40.47 | 43.10 | Virt. | 32.52 |
| LSB | | 57.13 | 63.15 | | 47.22 |

Table 4 shows the testing mIoU for *USA → Sing.*, *Day → Night*, and *A2D2 → Sem.* 2D-to-3D tasks. As can be seen, LSB consistently outperforms LSB (w/o $p_\phi$) and LSB (w/o $p_h$) across all the tasks, validating that aligning source and target feature in a common latent space helps effective transfer. Notably, LSB (w/o $p_\phi$) performs significantly worse, indicating that directly aligning features in the target feature space hinders segmentation learning.

## 5.4 SENSITIVITY ANALYSIS

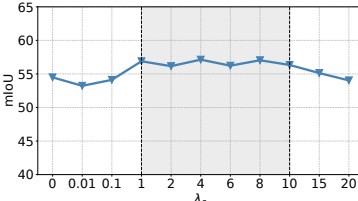 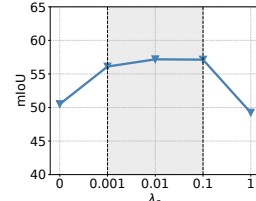 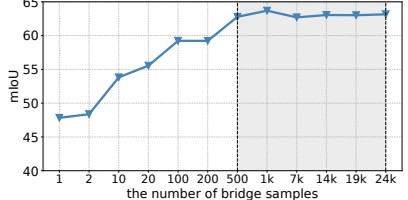

Figure 3: Effect of $\lambda_c$.  Figure 4: Effect of $\lambda_a$.  Figure 5: Effect of sample size.

**Effect of $\lambda_c$.** We conduct experiments on the 2D-to-3D task of *USA→Sing.* via the bridge domain *A2D2* to study the effect of $\lambda_c$. According to Figure 3, LSB is insensitive to a wide range of $\lambda_c \in [1, 10]$. Notably, LSB performs worse than LSB (w.o $\mathcal{L}_{con}^b$) when $\lambda_c < 0.1$, showing that a small $\lambda_c$ is not suitable for LSB to learn a consistent feature.

**Effect of $\lambda_a$.** To investigate the sensitivity of $\lambda_a$, we conduct the experiments on the 2D-to-3D task of *USA→Sing.* with the bridge domain *A2D2*. Figure 4 shows the testing mIoU w.r.t. $\lambda_a$. As can be seen, LSB achieves good performance in the range of $\lambda_a \in [0.001, 0.1]$. Moreover, increasing $\lambda_a$ can boost mIoU when $\lambda_a$ is small. However, excessively large values of $\lambda_a$ lead to a significant performance drop.

**Effect of the number of bridge samples.** We conduct experiments on the 2D-to-3D task of *Day→Night* with the bridge domain *A2D2* to study the effect of sample size in the bridge domain. As shown in Figure 5, mIoU increases with the number of bridge samples and stabilizes after 500 samples, suggesting that a relatively small number of bridge samples is sufficient for LSB to achieve stable performance.

## 6 CONCLUSION

In this paper, we introduce HMUDA, a novel setting designed to transfer knowledge between heterogeneous modalities using a bridge domain. For HMUDA, we propose a specialized framework LSB, with two distinct networks tailored for the source and target modalities. These networks are trained jointly with two segmentation losses specific to each network, alongside a feature consistency loss to promote similar feature representations between networks and a domain alignment loss to reduce domain discrepancies. Experimental results on various HMUDA benchmark datasets demonstrate the effectiveness of LSB in transferring knowledge across heterogeneous modalities. In our future work, we will apply LSB to other HMUDA tasks, such as image classification and object detection.

ETHICS STATEMENT

This research aims to transfer knowledge between different modalities. We exclusively used publicly available benchmark datasets (nuScenes, A2D2, SemanticKITTI, VirtualKITTI). Our work is intended for positive societal impact and does not develop inherently harmful technology.

REPRODUCIBILITY STATEMENT

To ensure reproducibility, we provide comprehensive details of our code, data, and experimental setup. The complete source code is included in the supplementary material, and all experiments are conducted on publicly available datasets. Further implementation specifics of LSB are described in Section 5.1.

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

## A   THE OVERALL ALGORITHM FOR THE LSB METHOD

---

**Algorithm 1** The LSB method.

---

**Require:** Source domain $\mathcal{S}$, target domain $\mathcal{T}$, bridge domain $\mathcal{B}$, learning rate $\eta$, hyper-parameters $\lambda_c$ and $\lambda_a$; trainable parameters $\boldsymbol{\theta}$: source network $\{h, f\}$, target network $\{\phi, g\}$ and projections $\{p_h, p_\phi\}$;

1: Initialize the source teacher model $\{\hat{h}, \hat{f}\}$.
2: **repeat**
3:     Sample a batch of data $\widetilde{\mathcal{S}} \subset \mathcal{S}, \widetilde{\mathcal{B}} \subset \mathcal{B}, \widetilde{\mathcal{T}} \subset \mathcal{T}$.
4:     **for** $(\mathbf{x}^s, \mathbf{y}^s) \in \widetilde{\mathcal{S}}$ **do**
5:         Compute loss $\mathcal{L}_{\text{seg}}^s(\mathbf{x}^s, \mathbf{y}^s)$ by Eq. (1).
6:     **end for**
7:     **for** $(\mathbf{x}^{bs}, \mathbf{x}^{bt}) \in \widetilde{\mathcal{B}}$ **do**
8:         Compute loss $\mathcal{L}_{\text{seg}}^b(\mathbf{x}^{bs}, \mathbf{x}^{bt})$ by Eq. (5).
9:         Compute loss $\mathcal{L}_{\text{con}}^b(\mathbf{x}^{bs}, \mathbf{x}^{bt})$ by Eq. (6).
10:     **end for**
11:     Compute loss $\mathcal{L}_{\text{ali}}(\widetilde{\mathcal{S}}, \widetilde{\mathcal{T}})$ by Eq. (9).
12:     Compute the overall loss $\mathcal{L}$ by Eq. (10).
13:     $\boldsymbol{\theta} \leftarrow \boldsymbol{\theta} - \eta\nabla_{\boldsymbol{\theta}}\mathcal{L}$;
14:     Update $\{\hat{h}, \hat{f}\}$ using EMA from $\{h, f\}$ by Eq. (2).
15: **until** convergence

---

## B   PROOF OF THEOREM 1

*Proof.* This proof relies on Theorem 2 in (Ben-David et al., 2010). We begin with the following inequality:

$$\mathcal{E}_{bs}(\{h, f\}) \leq \mathcal{E}_s(\{h, f\}) + \frac{1}{2}\big(d_{\mathcal{H}_s \Delta \mathcal{H}_s}(\mathcal{D}_s, \mathcal{D}_b)\big) + \lambda_s \tag{15}$$

and

$$\mathcal{E}_t(\{\phi, g\}) \leq \mathcal{E}_{bt}(\{\phi, g\}) + \frac{1}{2}\big(d_{\mathcal{H}_t \Delta \mathcal{H}_t}(\mathcal{D}_b, \mathcal{D}_t)\big) + \lambda_t$$

$$\leq \mathcal{E}_{bs}(\{h, f\}) + \mathcal{E}_{bt}(\{\phi, g\}) - \mathcal{E}_{bs}(\{h, f\}) + \frac{1}{2}\big(d_{\mathcal{H}_t \Delta \mathcal{H}_t}(\mathcal{D}_b, \mathcal{D}_t)\big) + \lambda_t$$

$$\leq \mathcal{E}_{bs}(\{h, f\}) + \big|\mathcal{E}_{bt}(\{\phi, g\}) - \mathcal{E}_{bs}(\{h, f\})\big| + \frac{1}{2}\big(d_{\mathcal{H}_t \Delta \mathcal{H}_t}(\mathcal{D}_b, \mathcal{D}_t)\big) + \lambda_t, \tag{16}$$

According to the modality assumption, we have:

$$\big|\mathcal{E}_{bt}(\{\phi, g\}) - \mathcal{E}_{bs}(\{h, f\})\big| \leq L\mathbb{E}_{(\mathbf{x}^{bs}, \mathbf{x}^{bt}) \sim \mathcal{D}_b}\big(d(h(\mathbf{x}^{bs}), \phi(\mathbf{x}^{bt}))\big) \tag{17}$$

By combining inequalities (15), (16) and (17), we obtain:

$$\mathcal{E}_t(\{\phi, g\}) \leq \mathcal{E}_{bs}(\{h, f\}) + \big|\mathcal{E}_{bt}(\{\phi, g\}) - \mathcal{E}_{bs}(\{h, f\})\big| + \frac{1}{2}\big(d_{\mathcal{H}_t \Delta \mathcal{H}_t}(\mathcal{D}_b, \mathcal{D}_t)\big) + \lambda_t$$

$$\leq \mathcal{E}_{bs}(\{h, f\}) + L\mathbb{E}_{(\mathbf{x}^{bs}, \mathbf{x}^{bt}) \sim \mathcal{D}_b}\big(d(h(\mathbf{x}^{bs}), \phi(\mathbf{x}^{bt}))\big) + \frac{1}{2}\big(d_{\mathcal{H}_t \Delta \mathcal{H}_t}(\mathcal{D}_b, \mathcal{D}_t)\big) + \lambda_t$$

$$\leq \mathcal{E}_s(\{h, f\}) + L\mathbb{E}_{(\mathbf{x}^{bs}, \mathbf{x}^{bt}) \sim \mathcal{D}_b}\big(d(h(\mathbf{x}^{bs}), \phi(\mathbf{x}^{bt}))\big) + (\lambda_s + \lambda_t)$$

$$+ \frac{1}{2}\big(d_{\mathcal{H}_s \Delta \mathcal{H}_s}(\mathcal{D}_s, \mathcal{D}_b)\big) + \frac{1}{2}\big(d_{\mathcal{H}_t \Delta \mathcal{H}_t}(\mathcal{D}_b, \mathcal{D}_t)\big), \tag{18}$$

$\square$

## C   DETAILED DATASETS SETTINGS

We construct eight HMUDA scenarios based on four datasets. Table 5 shows the details of each scenario.

| Scenario | $\mathcal{B}$ Train | | $\mathcal{S}$ Train | $\mathcal{T}$ Train | Val/Test |
|---|---|---|---|---|---|
| *USA → Sing.* | | | 15,695 | 9,665 | 2,770/2,929 |
| *Day → Night* | Sem. | 18,029 | 24,745 | 2,779 | 602/602 |
| *Virt. → A2D2* | | | 2,126 | 24,461 | 808/2,426 |
| *USA → Sing.* | | | 15,695 | 9,665 | 2,770/2,929 |
| *Day → Night* | A2D2 | 24,461 | 24,745 | 2,779 | 602/602 |
| *Virt. → Sem.* | | | 2,126 | 18,029 | 1,101/4,071 |
| *Sem. → A2D2* | Virt. | 2,126 | 18,029 | 24,461 | 808/2,426 |
| *A2D2 → Sem.* | | | 24,461 | 18,029 | 1,101/4,071 |

Table 5: The sample number in each split of datasets for all eight settings. Note that the training samples in target and bridge domains are without labels.

## D   EFFECT OF CROSS-MODAL ALIGNMENT

We study the effect of cross-modal domain alignment between the source and target domain on the HMUDA tasks. Instead of aligning the target domain with the source domain, we conduct an additional experiment, i.e., LSB (w. $\mathcal{L}_{ali}(\mathcal{B}, \mathcal{T})$), by minimizing the class centroid feature of the target network between the bridge and target domain. These class centroid features are computed using pseudo labels for the bridge and target

Table 6: Effect of cross-modal alignment on three 2D-to-3D HMUDA tasks.

| Method | $\mathcal{B}$ | USA → Sing. | Day → Night | $\mathcal{B}$ | A2D2 → Sem. |
|---|---|---|---|---|---|
| LSB (w. $\mathcal{L}_{ali}(\mathcal{B}, \mathcal{T})$) | A2D2 | 50.86 | 60.12 | Virt. | 44.44 |
| LSB | | 57.13 | 63.15 | | 47.22 |

domain samples. As shown in Table 6, LSB consistently outperforms LSB (w. $\mathcal{L}_{ali}(\mathcal{B}, \mathcal{T})$), indicating that aligning the target domain with the source domain is more effective.

## E   VISUALIZATION

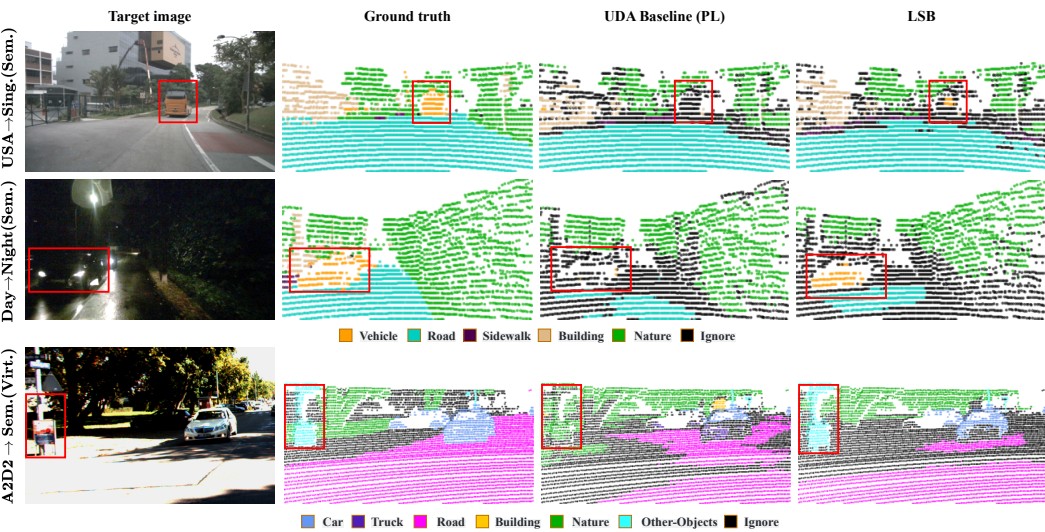

Figure 6: Qualitative results on three 2D-to-3D HMUDA tasks: *USA→Sing.*, *Day→Night*, and *A2D2→Sem.*. The (·) in the vertical axis denotes the bridge domain $\mathcal{B}$ used in the HMUDA task. For example, **USA→Sing.(Sem.)** denotes the transfer from *USA* to *Sing.* via the bridge domain *Sem.*.

**Qualitative results.** Figure 6 presents the qualitative segmentation results for various 2D-to-3D tasks, including *USA→Sing.*, *Day→Night*, and *A2D2→Sem.* As can be seen, the LSB method predicts

segmentation objects more accurately than the baseline method (PL). For instance, in the USA→Sing. task, buses are highlighted with red bounding boxes. We can see that the proposed LSB method correctly identifies the bus, while the baseline method PL misclassifies these points as 'Sidewalk'. Similar improvements are observed in the *Day→Night* and *A2D2→Sem* tasks, where LSB effectively recognizes 'Vehicle' and 'Other-Objects'.

**t-SNE Visualization.** Figure 7 presents the t-SNE visualization of target feature embeddings from the Source-Only model (a) and LSB (b) on the 2D-to-3D task of *Day→Night* via the bridge domain *A2D2*. Compared to Source-Only, LSB shows improved clustering for the vehicle class, with more compact and better-separated features across domains.

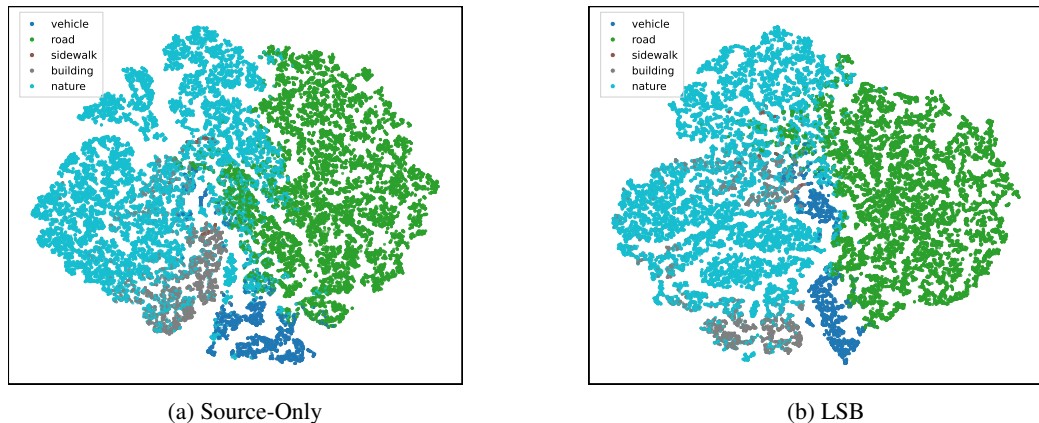

(a) Source-Only  (b) LSB

Figure 7: t-SNE visualization of target feature embeddings for (a) Source-Only and (b) LSB.

## F  LIMITATIONS

This work focuses on establishing a novel unsupervised domain adaptation framework for heterogeneous modalities, while in application, we only evaluate the vision-based modalities, i.e., 2D images and 3D point clouds. In our future work, we will apply LSB to other HMUDA tasks, such as image classification and object detection.

## G  LARGE LANGUAGE MODEL USAGE STATEMENT

This work did not use large language models (LLMs) for any part of the research methodology, data analysis, or conceptual development. LLMs were used solely during the writing and revision stages to improve the clarity, grammar, and expressiveness of the manuscript. Specifically, the model was employed as a language editing tool to enhance sentence fluency and readability. All technical content, including methods, experiments, results, and interpretations, were developed entirely by the authors.

