# OpenReview forum: "Heterogeneous-Modal Unsupervised Domain Adaptation via Latent Space Bridging"
_ICLR.cc/2026/Conference — ICLR 2026 Conference Withdrawn Submission_

### Official Review · Reviewer_DhMP · 2025-10-28

**Soundness:** 2
**Presentation:** 2
**Contribution:** 2
**Rating:** 4
**Confidence:** 4

**Summary:**

The paper proposes a Heterogeneous UDA method that aims to transfer knowledge across two different modalities from a labeled source domain to an unlabeled target domain. To facilitate cross-modal knowledge transfer, an additional bridge domain is introduced, which is assumed to contain paired data from both modalities of the same input. A consistency loss and an alignment loss are proposed to guide the dual-network architecture in learning intra-domain discriminative representations while reducing inter-domain discrepancies. Extensive experiments on multiple 2D–3D segmentation UDA benchmarks demonstrate the effectiveness of the proposed approach.

**Strengths:**

+ In real-world scenarios, labeling 3D data is significantly more difficult and costly than labeling 2D images. From this perspective, 2D–3D UDA represents a practical and meaningful task.

+ The authors conduct thorough ablation studies to validate the effectiveness of the proposed modules and the chosen hyperparameters.

+ The paper is well written and easy to follow.

**Weaknesses:**

- The paper claims that 2D–3D UDA with a bridge domain is a new setting. However, the proposed bridge domain introduces a strict limitation, as it requires data from both modalities corresponding to the same input. Such an assumption makes data collection expensive and impractical, thereby weakening the claimed novelty and limiting the setting’s applicability to generalized UDA tasks.

- In the experimental section, all results are limited to 2D–3D modalities. The work would be more meaningful if the authors could extend the setting to other modality pairs, such as visible–infrared.

- 2D–3D UDA for segmentation has already been explored in several prior studies. The authors should cite these works and compare their method against them, such as:

[1] Zhang, Yachao, et al. "Self-supervised exclusive learning for 3d segmentation with cross-modal unsupervised domain adaptation." Proceedings of the 30th ACM International Conference on Multimedia. 2022.

[2] Spoecklberger, Johannes, et al. "Exploring Modality Guidance to Enhance VFM-based Feature Fusion for UDA in 3D Semantic Segmentation." Proceedings of the Computer Vision and Pattern Recognition Conference. 2025.

[3] Wu, Yao, et al. "Unidseg: Unified cross-domain 3d semantic segmentation via visual foundation models prior." Advances in Neural Information Processing Systems 37 (2024)

[4] Cardace, Adriano, et al. "Exploiting the complementarity of 2d and 3d networks to address domain-shift in 3d semantic segmentation." Proceedings of the IEEE/CVF Conference on Computer Vision and Pattern Recognition. 2023.


- The proposed dual-network structure lacks novelty, as similar architectures have been widely adopted in prior 2D–3D UDA studies [4]. Moreover, using EMA to stabilize knowledge transfer between networks has also been extensively explored in previous works [5][6]. The proposed consistency loss is essentially an MSE loss with a simple regularization term, offering limited methodological innovation.

[5] Huo, Xinyue, et al. "Focus on your target: A dual teacher-student framework for domain-adaptive semantic segmentation." Proceedings of the IEEE/CVF International Conference on Computer Vision. 2023.

[6] Ge, Yixiao, Dapeng Chen, and Hongsheng Li. "Mutual mean-teaching: Pseudo label refinery for unsupervised domain adaptation on person re-identification." ICLR (2020).

**Questions:**

- Table 3 shows that the UDA performance varies across different bridge domains. The authors should provide a more in-depth analysis of how to select an appropriate bridge domain. For instance, do data volume and the distribution gap between the bridge domain and the source/target domains affect the results?

- The proposed method is mainly designed for cross-modal transfer. It would be interesting to investigate how the method performs when the source and target domains each consist of multiple datasets within the same modality, i.e., whether intra-modality domain differences could lead to performance degradation: S = {(x^{s_1}, y_{s_1}), (x^{s_2}, y_{s_2})}, T = {x^{t_1}, x^{t_2}}.
Since autonomous driving data are typically collected under diverse driving conditions, it is difficult to ensure that all samples originate from the same domain. Including such an analysis would further strengthen the paper and highlight the robustness of the proposed approach.

---

### Official Review · Reviewer_EY3G · 2025-10-29

**Soundness:** 2
**Presentation:** 3
**Contribution:** 2
**Rating:** 2
**Confidence:** 5

**Summary:**

This paper introduces a new setting: Heterogeneous-Modal Unsupervised Domain Adaptation, aiming to transfer knowledge between heterogeneous modalities. For this setting, a framework LSB is proposed, which contains two distinct networks tailored for the source and target modalities. LSB is trained jointly with two segmentation losses.

**Strengths:**

- This paper introduces an interesting task: Heterogeneous-Modal Unsupervised Domain Adaptation (HMUDA).
- The authors provide detailed ablation studies to analyze the effectiveness of the proposed components.
- The method is simple and transudative, and the described algorithmic process is intuitive.

**Weaknesses:**

- In terms of technical innovation, the two components proposed in this paper, namely a feature consistency loss and a domain alignment loss, are common techniques. They are similar to the cross-modal distillation loss and class prototype loss in prior work, respectively, and have been used multiple times.
- The concept of the “Bridge domain” is not clear. The source of the image and point cloud data for the Bridge domain is not clearly described in the paper. The authors need to provide further explanation on this point.
- In terms of experimental comparison, although the authors have proposed a new task (HMUDA), it is necessary for them to try recent multi-modal domain adaptation methods (published in the last two years) to validate their effectiveness on this task.

**Questions:**

The author needs to respond to the shortcomings of the content.

---

### Official Review · Reviewer_DAdU · 2025-10-31

**Soundness:** 3
**Presentation:** 3
**Contribution:** 1
**Rating:** 6
**Confidence:** 3

**Summary:**

Given a labeled 2D image set $\mathcal{S}$, an unlabeled image–point cloud pair $\mathcal{B}$ from a different domain, and an unlabeled point cloud $\mathcal{T}$, the paper proposes a method to transfer the knowledge of 𝒮 to 𝒯 through a latent space bridge between the image and point cloud modalities.

**Strengths:**

- The proposed framework has the advantage of being applicable not only to vision-based modalities such as images and 3D point clouds but also to other modalities across different domains through the same bridging mechanism.
- Unlike the xMUDA family of methods, the paper effectively addresses scenarios where the target domain contains only unlabeled single-modality data.

**Weaknesses:**

- The proposed method appears to rely heavily on the quantity and accuracy of the bridge-domain data. Compared to prior approaches that utilize pretrained VLMs such as CLIP for UDA/SFDA, the practical advantages and distinctions of this method should be more clearly emphasized.
- Although proposing the HMUDA method is meaningful, it seems that the approach is not universally applicable to all multi-modal UDA tasks. Rather, it is limited to cases where an explicit bridge domain exists between two modalities.
- Typographical issues
   - Inconsistent usage of same word (hyperparameter, hyper-parameter)
   - Line 375: “hyper-paramters”
   - Line 435: “centriod”
   - Line 151: “a unlabeled”

**Questions:**

It would be beneficial for the paper to emphasize the broader applicability of the proposed bridge domain beyond vision-based modalities such as images and 3D point clouds. While existing multimodal pretrained models already address cross-modal learning, the paper should more clearly highlight the unique advantages and contributions of the proposed approach in comparison.

---

### Official Review · Reviewer_8K3f · 2025-10-31

**Soundness:** 2
**Presentation:** 3
**Contribution:** 2
**Rating:** 4
**Confidence:** 3

**Summary:**

The authors introduce a new approach to Heterogeneous Multi-modal Domain Adaptation, addressing a gap in existing methods where the source and target domains differ in modality, but labeled data are available only in the source domain.
Their approach leverages a bridge domain that contains both modalities but no labels, serving as a link between the source and target domains during training.
After presenting their method, the authors provide a theoretical analysis of its convergence and a comprehensive experimental evaluation.

**Strengths:**

- First, the paper is well-written and easy to follow from beginning to end.
- It addresses an interesting problem in the domain adaptation literature, which is not resolved, and shows good results compared to other competitors.
- In addition, the authors provide a theoretical analysis of the convergence of their method.

**Weaknesses:**

- The authors claim that they are the first paper to study heterogeneous unsupervised domain adaptation, but different methods tackle this problem before, like the xMUDA that you compare with. A clearer positioning of the paper in the literature is needed to enhance the contribution's clarity.

-Although this experimental setup has not been explored before, I see a limitation in the requirement for a third domain containing both modalities. It is not evident that such a setup would be easier to obtain in real-world scenarios. In practice, it may be more feasible to annotate a few target samples to achieve a heterogeneous DA setting or to utilize source domains that already encompass both modalities.

-The proposed method also relies on numerous loss functions and specific architectural components (such as linear projections). This raises two concerns: how to effectively validate all the hyperparameters (see questions), and how easily the approach can be adapted to other tasks—a limitation that the authors themselves acknowledge.

**Questions:**

- How did you validate the hyperparameters?
- Have you tried implementing heterogeneous domain adaptation methods that use a few labeled samples in the target domain? As mentioned in the weaknesses, your setup appears somewhat restrictive, as it requires an additional domain that contains both modalities. From a practical standpoint, it might be faster for practitioners to annotate a small number of target samples instead. I am also curious about how standard heterogeneous DA methods would perform on your dataset.
- Regarding the alignment loss, could you elaborate on your choice? You opted for a centroid-based loss, but alternatives such as correlation alignment or optimal transport losses could also be considered. What motivated your specific selection?

---

### Note · Authors · 2025-11-14

I have read and agree with the venue's withdrawal policy on behalf of myself and my co-authors.